GFVO: the Genomic Feature and Variation Ontology

Baran Joachim 1 kim@codamono.com
Durgahee Bibi Sehnaaz Begum 2
Eilbeck Karen 2
Antezana Erick 3
Hoehndorf Robert 4
Dumontier Michel 1
1 Stanford Center for Biomedical Informatics Research, School of Medicine, Stanford University , Stanford, CA , USA
2 Department of Biomedical Informatics, School of Medicine, University of Utah , Salt Lake City, UT , USA
3 Department of Biology, Norwegian University of Science and Technology , Trondheim , Norway
4 Computer, Electrical and Mathematical Sciences & Engineering Division and Computational Bioscience Research Center, King Abdullah University of Science and Technology , Thuwal , Kingdom of Saudi Arabia
Vision Todd
Electronic publication date: 2015 May 5
Publication date: 2015
Volume: 3
Electronic Location ID: e933
Received 2014 Nov 4; Accepted 2015 Apr 14
Copyright: © 2015 Baran et al.
Copyright year: 2015
Copyright holder: Baran et al.
License: This is an open access article distributed under the terms of the Creative Commons Attribution License, which permits unrestricted use, distribution, reproduction and adaptation in any medium and for any purpose provided that it is properly attributed. For attribution, the original author(s), title, publication source (PeerJ) and either DOI or URL of the article must be cited.
License URL: https://creativecommons.org/licenses/by/4.0/

Keywords: Bioinformatics, Genomics, Ontology

Funding: National Human Genome Research Institute R01HG004341 Schlumberger Foundation Faculty for the Future This work was conducted using the Protégé resource, which is supported by grant GM10331601 from the National Institute of General Medical Sciences of the United States National Institutes of Health. Karen Eilbeck was funded by National Human Genome Research Institute R01HG004341, and Bibi Sehnaaz Begum Durgahee was funded by the Schlumberger Foundation Faculty for the Future. No other funding was received that directly supported the development of the ontology.

==============================
Falling costs in genomic laboratory experiments have led to a steady increase of genomic feature and variation data. Multiple genomic data formats exist for sharing these data, and whilst they are similar, they are addressing slightly different data viewpoints and are consequently not fully compatible with each other. The fragmentation of data format specifications makes it hard to integrate and interpret data for further analysis with information from multiple data providers. As a solution, a new ontology is presented here for annotating and representing genomic feature and variation dataset contents. The Genomic Feature and Variation Ontology (GFVO) specifically addresses genomic data as it is regularly shared using the GFF3 (incl. FASTA), GTF, GVF and VCF file formats. GFVO simplifies data integration and enables linking of genomic annotations across datasets through common semantics of genomic types and relations.

Availability and implementation. The latest stable release of the ontology is available via its base URI; previous and development versions are available at the ontology’s GitHub repository: https://github.com/BioInterchange/Ontologies; versions of the ontology are indexed through BioPortal (without external class-/property-equivalences due to BioPortal release 4.10 limitations); examples and reference documentation is provided on a separate web-page: http://www.biointerchange.org/ontologies.html. GFVO version 1.0.2 is licensed under the CC0 1.0 Universal license (https://creativecommons.org/publicdomain/zero/1.0) and therefore de facto within the public domain; the ontology can be appropriated without attribution for commercial and non-commercial use.

Introduction

As the cost of genomic laboratory experiments has fallen, vast amounts of genomic data are produced by high-throughput sequencing technologies as well as inexpensive microarrays in a wide range of studies (cf. Grada & Weinbrecht, 2013; Guarnaccia et al., 2014). Large cancer genome repositories, e.g., the 12,807 cancer genomes sequenced by institutes of the International Cancer Genome Consortium (ICGC, release 18, http://www.icgc.org), contain extensive data about structural variations of DNA and proteins. Epigenetic cohort studies contribute substantially to the scientific knowledge base too, with millions of data points being provided by recent studies (Walker et al., 2011; Lam et al., 2012). Whilst the produced data are rich in information content, they are to varying degree kept in data silos due to the lack of a single unified standard for data exchange and data integration. This increases the cost of further data analysis, leaving much of that data not adequately used or simply unused (Sboner et al., 2011).

Genomic data is at the moment predominantly represented as plain text- or binary-data. Nucleotide or peptide sequences are expressible via the broadly known FASTA file format (Pearson & Lipman, 1988; included in the GFF3 format), sequencing alignments/maps can be encoded as SAM/BAM files (Li et al., 2009), genomic features of many kinds can be captured via GFF3 (http://www.sequenceontology.org/gff3.shtml) or its predecessor GTF (http://mblab.wustl.edu/GTF2.html); structural variations of DNA can be captured using VCF (Danecek et al., 2011) or by the extended GFF3 file format GVF (Reese et al., 2010). Each of these standards is specified using manifests written in plain English, and thus, by and large are neither machine-interpretable—as opposed to merely machine-readable—nor formally precise, which leaves room for speculation about the encoding/representation of some data.

Markup-related file formats, such as the eXtensible Markup Language (XML; http://www.w3.org/TR/2006/REC-xml11-20060816), provide a mitigated approach that maintains a human-readable text file format, whilst at the same time offering rudimentary support for a formal data format specification via Document Type Definitions or XML Schema (for a comparison between the two standards see Bex, Neven & Den Bussche, 2004). The Systems Biology Markup Language (SBML), (Hucka et al., 2003), is an example of an XML-based format for representing computational models and biochemical networks. A similar approach towards representing protein–protein interaction networks culminated in the data representation Proteomics Standards Initiative Molecular Interaction XML format (PSI-MI), using an XML Schema for data validation (Hermjakob et al., 2004). A comparison among SBML, PSI-MI and the BioPAX language, (Strömbäck & Lambrix, 2005), highlighted the advantages of BioPAX’s approach regarding data integration and linked data. BioPAX, (Demir et al., 2010), is a specification formalized in the Web Ontology Language (OWL; Antoniou & Harmelen, 2009), which facilitates data integration and data linkage using the Resource Description Framework (RDF; http://www.w3.org/TR/2014/REC-rdf11-concepts-20140225).

Some benefits of the Semantic Web technologies, such as RDF and OWL, have also been highlighted in other fields. The Semantic Web has been suggested to enrich academic publications with software interpretable annotations, (Shotton et al., 2009), a semantic linked data network of drug targets has been presented as a data integration solution of heterogeneous drug related data sets in Chen, Ding & Wild (2012), and a broader discussion on the role of the Semantic Web technologies in life sciences was given in Antezana, Kuiper & Mironov (2009).

Generic feature and genomic variation file formats

GFF3 is a cornerstone in many GMOD tools, (http://www.gmod.org), such as the generic genome browser GBrowse (Stein et al., 2002), it is input to genomic variant annotation tools such as ANNOVAR, (Wang, Li & Hakonarson, 2010), and it is used to format results of genome annotation pipelines such as MAKER, (Cantarel et al., 2008). The GFF3 specification also includes the FASTA file format specification, since FASTA file contents are permitted in the trailing part of GFF3 files. Genomic variations are represented using either GVF or VCF file formats, which are utilized interchangeably in projects of the European Bioinformatics Institute’s (EBI; Lappalainen et al., 2013), the 1,000 Genomes Project as well as the Broad Institute (The 1000 Genomes Project Consortium, 2012), among other data providers.

GFF3-, GTF-, GVF-, and VCF-specifications define tab-delimited text file formats, which are instances of flat file formats. For all specifications, the text file contents are of variable column-length and additional predefined delimiters (such as a semicolon) are being used for further value separation within certain columns. Multiple Perl-inspired (http://perldoc.perl.org/index-pragmas.html) “pragma” statements are typically declared in the beginning of the files to further determine the interpretation of genomic data contents below. Genomic data are either annotated in predefined columns (tab-delimited; column position determines data content), where some columns are allowed to contain multiple tag/value pairs, or, other composite or ordered data structures (e.g., lists). Figure 1 shows an example of a GFF3 file and the implicit relationships between data within it.

Figure 1 A GFF3 file example that shows the implicit relationships between data in the file and its genomic locus interpretation.

From flat files to a machine interpretable data format

Differences in the genomic feature and variation file formats require multiple software implementations for reading their data files, where data integration across file formats is difficult due to disparities in data labeling and data encoding. The goal of this paper is to overcome these issues by introducing the Genomic Feature and Variation Ontology (GFVO), which addresses this status quo by providing well described named ontological concepts and relations, by defining formal constraints on them, and by potentially enabling unified data access through a generic data representation.

GFVO predominantly unifies naming conventions and data encoding by introducing unique concepts for genomic features and variation representation. For example, the GFVO class “SequenceVariant” expresses sequence variants that are encoded using the “Variant_seq” attribute in GVF files and as either “ALT” column or “ALT” information field in VCF files. GFVO also resolves encoding differences between genomic feature and variation file formats. For example, gametic phase is encoded very differently in GVF and VCF file formats. GVF denotes gametic phase with its “Genotype” attribute in conjunction with “Phased” attribute, or alternatively, “##phased-genotypes” pragma statement. VCF denotes gametic phase in its “GT” additional information field using the special characters “∣” (phased) and “/” (unphased). In GFVO, gametic phase is represented via the unique class “GameticPhase” that encodes both GVF and VCF use cases.

Secondarily, GFVO enables easier data processing and data integration due to the use of World Wide Web Consortium (W3C) backed data standards. Data represented by GFVO can be expressed in one of the many interchangeable Semantic Web file formats (s.a. RDF N-Triples, RDF Turtle, or JSON-LD). Data in those formats can be parsed in many popular programming languages, or, it can be directly loaded into database management systems without requiring further data transformation. For example, RDF Turtle can directly be imported into Virtuoso and other triple stores, while the newer JSON-LD can be imported into MongoDB, Elasticsearch, and other NoSQL database management systems.

Methods

GFVO is modeled in OWL and made available in RDF/XML format (http://www.w3.org/TR/rdf-syntax-grammar). The ontology’s expressiveness is within SRIQ with datatype properties, which is a Description Logic (Baader et al., 2003) fragment whose development has previously been motivated by application in medical terminology (Horrocks, Kutz & Sattler, 2005). Table 1 gives an overview of the number of classes, properties, link-outs, and restrictions that are defined in GFVO.

Table 1 Overview of classes and properties in GFVO.

Total number of classes and properties in GFVO, number of classes/properties that have equivalences to SIO, number of classes based on GFF3, GTF, GVF and VCF specifications (not mutually exclusive), number of classes with reference to Wikipedia articles, disjointness axioms, and property restrictions.

	Total number	Equivalence with SIO	Equivalence with SO	
Classes	102	40	13	
…modeled from GFF3	23	11	3	
…modeled from GTF	13	11	2	
…modeled from GVF	62	23	12	
…modeled from VCF	42	15	7	
Class metadata				
…Wikipedia references	53	n/a	n/a	
…pairwise disjoint auxioms	6	n/a	n/a	
…disjoint collection auxioms	13	n/a	n/a	
…with property restrictions	32	n/a	n/a	
Datatype properties	1	1	0	
Object properties	32	31	0	

Ontology design

GFVO was designed to mirror the class and object property hierarchies of the Semanticscience Integrated Ontology (SIO; Dumontier et al., 2014) release version 1.0.10. SIO is an ontology of basic types and relationships for biomedical knowledge discovery that serves as an upper-level ontology to GFVO. The term upper-level ontology has been defined as a highly abstracted ontology that does not refer to identifiable and/or concrete entities of a domain, but also as an ontology that can be used to impose structure on a lower-level ontology as well as provide relations to other ontologies (Musen, 2013). SIO adheres to the latter by ensuring semantic interoperability via relations between SIO classes as well as properties and the Basic Formal Ontology (BFO; Grenon, Smith & Goldberg, 2004), Relation Ontology (RO; Smith et al., 2005), and the Phenotypic Quality Ontology (PATO) (http://obofoundry.org/wiki/index.php/PATO:Main_Page). Formal relationships between GFVO’s and SIO’s classes and properties are included in GFVO via “owl:equivalentProperty” and “owl:equivalentClass” OWL-properties respectively.

Terminological requirements to describe data contents as covered by the GFF3 version 1.21, GTF version 2, GVF version 1.07, and VCF version 4.1 & 4.2 specifications were gathered by tabulating all named data elements within the specifications. Terminology from specifications was preserved wherever possible, unless a single unifying term seemed more appropriate. This design decision results in pre-coordinated GFVO classes. Pre-coordinated classes are understood as the combination of atomic concepts, e.g., GO accession “GO:0006260” that is labeled as “DNA replication,” (Ashburner et al., 2000), whereas post-coordinated classes are interpreted as the combination of atomic classes via relations and axioms. The widely utilized SNOMED CT uses pre-coordinated terminology too, where post-coordination has been shown to have possible negative effects in terms of misinterpretation of composed SNOMED CT classes (Rector & Iannone, 2012). Pre- and post-coordination have been studied in biomedical sciences in general and it has been found that pre-coordinated ontologies require less user guidance (Schulz et al., 2010). Table 2 provides an overview of the number of GFVO classes that are representing data as it can be found in the genomics feature and variation file formats.

Table 2 Overview of file-format data structures captured by GFVO.

Overview of the number of data structures (columns, key/value pairs, other) in genomics file formats that are captured by GFVO and other ontologies/frameworks.

Specification	Fixed columns	Feature attributes and key/value properties	Pragma statements and information fields	
Table column description	Columns with a single data type	Boolean attributes and key/value properties directly associated with genomic features	Other information that is stated in the header or outside of the scope of genomic feature annotations (e.g., comments or FASTA appendix)	
	Entities identified in specification and represented by classes in GFVO; entities represented in other ontologies/frameworks as either class or property are noted in brackets	
GFF3	8 (+1 SO)	5	5 (+4 RDF schema)	
GTF	9	n/a	6	
GVF	8 (+1 SO)	25	27 (+4 RDF schema)	
VCF	6	24	15	

Naming conventions in GFVO are requiring camel case rules for all identifiers, which have to be accompanied by an English-language label for classes and properties (“rdfs:label”). The camel case for class identifiers starts with an upper case letter (e.g., identifier “CodingFrameOffset” with “Coding Frame Offset” as “rdfs:label”). Object- and datatype-properties start with a lower case letter (e.g., identifier “isPartOf” with “is part of” as “rdfs:label”). Identifiers and labels in GFVO were chosen as close as possible to the genomics file format specifications to assist with the adoption of the ontology by bioinformaticians and life science researches.

Similar to SIO, only one datatype property was modeled for representing literal values. Domain- and range-restrictions have been placed as machine interpretable reasoning support that also serve as application guide. Designated object properties have defined range restrictions to the Sequence Ontology (SO; Eilbeck et al., 2005), Variation Ontology (VariO; Vihinen, 2013) and Feature Annotation Location Description Ontology (FALDO; Bolleman et al., 2014, pre-print). The utilization of existing ontologies for specific data representation needs facilitates data integration among genomic data resources through ontology reuse. Examples are provided as Supplemental Information that show the intended use of GFVO including the use of the aforementioned ontologies in conjunction with GFVO.

Descriptive explanations about GFVO’s 102 classes, 33 properties and their intended usage have been provided in comment sections (“rdfs:comment”; cf. tutorial by Horridge et al., 2007). It is shown in Table 3 that the comment sections in GFVO add substantial documentation in comparison to FALDO, SIO, SO, and VariO. Tables 4 and 5 in Supplemental Information break down detailed mappings between the genomics specifications and GFVO, FALDO, SO, and RDF Schema; Table 4 maps every GFVO class to data content in the genomic file formats, whereas Table 5 shows use of FALDO, SO, and RDF Schema. Where applicable, GFVO references Wikipedia (“rdfs:seeAlso”) to provide extra information about the concept that classes encapsulate. References to GFF3, GTF, GVF, and VCF specifications (“rdfs:isDefinedBy”) are denoting the modeling origin as well as applicability of classes to the respective file formats; 1 the instantiation of classes is not restricted by this annotation though and it is encouraged to make use of GFVO classes according to data representation needs.

Table 3 Coverage of class documentation in terms of word- and class-counts.

Coverage of class documentation in genomics related ontologies. Coverage is denoted by total number of words of documentation as well as on a normalized per-class basis.

Ontology	Total number of words in comments/descriptions	Total number of classes	Average number of words in descriptions per class	
			average (minimum; median; maximum)	
FALDO 31st November 2014	477	18	26.5 (6; 25; 72)	
GFVO 1.0.2	4,478	102	43.9 (12; 38.5; 133)	
SIO 1.0.10	23,412	1,414	16.56 (5; 16; 73)	
SO 1.329	7,296	2,254	3.24 (1; 13; 97)	
VariO 16th January 2014	4,612	384	12.01 (1; 8; 74)	

Ontology testing

Protégé 4.3 (http://protege.stanford.edu) with HermiT 1.3.7 reasoner plug-in were used to test whether GFVO is free of inconsistencies and unsatisfiable classes—within the reasoning boundaries of HermiT. No inconsistencies were highlighted by the reasoner and inferred class and property hierarchies have been manually evaluated and it has been determined that they follow the intended design.

Examples were created for selected use cases, which also serve as test bed for verifying the ontology’s expressiveness. Examples 1, 2, 3, 3a, 4, and 5 in Supplemental Information show a possible solution for encoding genes and their transcription factor binding sites, sequence alignments, phased and unphased genotypes, sequence variants, and Phred quality scores, and data filtering/annotating, respectively. All examples were inspected for errors using Protege/HermiT; no errors were shown regarding the RDF Turtle format and the example are inconsistency free.

Results

GFVO unifies the vocabulary of the genomic feature and variation file format specifications by providing a well-defined and documented set of classes and properties that capture the gist of genomic data. Designated classes classes (e.g., “AlleleCount,” “Zygosity,” “FragmentReadPlatform”) are provided for concepts that are only indicated by special and varied encodings in the file format specifications. Character based data encoding of the genomic feature and variation file formats, s.a. “∣” for phased genotypes in the VCF specification, have been replaced by declarative classes with extensive documentation in GFVO.

Representing four genomic feature and variation file formats using GFVO enables easier access to the understanding and interpretation of genomic data. Independent of the file format specific encoding and formatting of the GFF3 (incl. FASTA), GTF, GVF, and VCF file formats, data represented using GFVO provides a single point of access for data interpretation due to its well-documented class hierarchy. Genomic data represented using GFVO is also machine interpretable due to the adherence of Semantic Web standards as defined by the W3C.

Discussion

This work identified and formalized classes and relations in the genomics feature and variation file formats GFF3, GTF, GVF, and VCF. GFVO consolidates encodings across these file formats with distinct and machine interpretable concepts. It has been demonstrated by examples that GFVO is suitable for describing the data contents of common genomic feature and genomic variant file formats.

Encoding genomic features and variations using GFVO comes at the cost of increased data size. This is partially due to the descriptive names that are featured in GFVO. For example, the representation of VCF allele counts (“AC” additional information) via the GFVO class “AlleleCount.” Exclusive use of classes to represent genomic data is another contributing factor towards size, since each genomic datum needs to be encoded as a class instance. Use of multiple datatype properties could alleviate the size increase, but this was not implemented to stay in line with SIO design decisions. Size comparisons between GFF3-, GTF-, GVF-, and VCF-files and their Semantic Web representation fall beyond the scope of this paper; size requirements for Semantic Web file formats vary greatly between standards, some of which permit omission of repetitive information (s.a. RDF Turtle and JSON-LD) whilst other standards require the explicit encoding of all data (s.a. RDF N-Triples).

The ontology’s distribution under the CC0 1.0 Universal license allows for adoption in commercial and non-commercial projects without the need to cite, to attribute, or otherwise reference the ontology, this paper, or its licensing terms in any form. GFVO is therefore de-facto within the public domain.

Conclusion

An ontology—GFVO—for describing the file contents of GFF3, GTF, GVF, and VCF files was introduced. GFVO builds on the existing ontologies FALDO, SIO, and SO. It was demonstrated that the ontology captures a set of use cases for describing genomic data; supporting evidence was given that indicates that GFVO captures data of the genomics feature and variation formats completely.

Supplemental Information

Supplemental Information 1 Supplementary Information

Click here for additional data file.

Supplemental Information 2 Mapping of GFVO classes to GFF3, GTF, GVF, and VCF data structures

GFVO classes mapped to data structures as they are defined in the GFF3, GTF, GVF, and VCF specifications (columns, key/value type pairs, etc.).

Click here for additional data file.

Supplemental Information 3 Other ontology classes and frameworks mapped to GFF3, GTF, GVF, and VCF data structures

Use and re-use of other ontology classes and frameworks to GFF3, GTF, GVF, and VCF data structures besides the utilization of GFVO classes.

Click here for additional data file.

Paul Avillach reviewed and provided feedback on an earlier version of the manuscript that presented two ontologies specific to the GFF3 and GVF file formats; Chris Mungall provided feedback regarding the GFF3 specific ontology as well. Raoul Jean Pierre Bonnal, Toshiaki Katayama, Francesco Strozzi brought forward suggestions to improve GFVO’s practical applications. Takatomo Fujisawa corrected bugs in the late development stages of GFVO.

The organizers of BioHackathon and their supporting bodies—the Integrated Database Project (Ministry of Education, Culture, Sports, Science and Technology of Japan), the National Bioscience Database Center (NBDC), and the Database Center for Life Science (DBCLS)—enabled the conception of the ontology.

Additional Information and Declarations

Competing Interests

Author Contributions

1 The VCF specification is no longer maintained by the 1,000 Genomes Project, but instead is maintained by the Global Alliance for Genomics and Health Data Working group file format team now. For backwards compatibility, GFVO includes links to the old as well as new location of the VCF 4.1 specification; the ontology also includes links to the current location of the VCF 4.2 specification.

The authors declare there are no competing interests.

Joachim Baran analyzed the data, contributed reagents/materials/analysis tools, wrote the paper, prepared figures and/or tables, reviewed drafts of the paper, created and developed the ontology.

Bibi Sehnaaz Begum Durgahee and Erick Antezana reviewed drafts of the paper, contributed to the development of the ontology by making active changes to it.

Karen Eilbeck reviewed drafts of the paper, contributed to the development of the ontology with her expertise in genomics file formats.

Robert Hoehndorf reviewed drafts of the paper, contributed to the development of the ontology by providing insights on his work on semantic representations of GFF3 files.

Michel Dumontier reviewed drafts of the paper, contributed to the development of the ontology with support regarding the upper ontology SIO.

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
