# Peer review of "GFVO: the Genomic Feature and Variation Ontology"

_PeerJ, doi:10.7717/peerj.933_

## Round 0.1 · original submission · Major Revisions

· Academic Editor

Major Revisions

The ontology itself appears to be a valuable contribution and expertly designed. However, PeerJ only publishes research papers, and so the addition of a Results section, with some demonstration of the utility of the ontology, as per Reviewer #2, will be needed for publication in this journal. Reviewers #2 and #3 also point out areas where additional context would improve the paper and make the intended uses for the ontology (either through improving existing practices or making possible new ones) clearer to the journal's readership in biological/health/medical research. Please also be sure to address the specific comments by Reviewer #2 under 'Validity of Findings'.

·

Basic reporting

No Comments

Experimental design

The ontology described in this paper is sorely needed by the research community, and based on the examples provided in the supplemental materials, it is sufficiently expressive to do what it is intended to do. Of course, the real trick is getting the community to adopt something like this. Adoption will depend on software implementations that hopefully will come next.

Validity of the findings

Again, well done; the ontology is available from a public web site and while I am not an ontology expert, the content of GVFO certainly "makes sense" to me.

Additional comments

This is a really well written paper; it is rare for me to review a paper that doesn't have at least something missing or poorly written sections.

Reviewer 2 ·

Basic reporting

There is not a clear purpose for the need for such an ontology. There is a brief statement at the end of the abstract, that it “simplifies data integration and enables linking of genomic annotations across datasets”, but this was not demonstrated in the contents of the article itself. If the idea is that the purpose of the ontology is for file format conversion or I/O, then that should be suggested as a motivation early to justify the work, and explained why current file format conversion tools are not satisfactory, and why an ontology is the necessary way to solve the problem (as opposed to a database schema, taxonomy, etc).

One way to demonstrate the utility, I would recommend that the authors come up with a set of related examples for features that are best encoded in each file format, which would simulate the results of a variety of independent analyses. Then, show how all 4 example features could then be linked with GFVO, and how the ontology structure enables reasoned results that would not otherwise be achievable.

*I am not sure what the section entitled “Generic Feature and Genomic Variation File Formats” is doing where it is. Is it it’s own section? Is it a sub-section of the Intro? If the latter, it is poor writing style to have a single sub-section.

*no links to the Git repo (it lists that it is available from GitHub, which it was as I found it through google, but the link nor the repo name was not in the manuscript.)

*I do not find Table 3 of much interest or use, nor as a valid measure of "performance" as suggested in the text (see line 113).

*There is no legend for tables 4 & 5.

Experimental design

The examples listed in the supplementary information are useful, especially the contained documentation elements; however, since the premise of the ontology is the encoding of various elements of from different file formats, each example cannot be interpreted without the corresponding source data in it's native format. Furthermore, a cartoon diagram explaining how the example elements map to the genomic feature(s) would make them much easier to understand. If it is too much to include the source formats in addition to the turtle data, then they should be made available at the git repository for comparison.

Validity of the findings

*why aren’t there more links to external ontologies? There are some very obvious classes and properties that are not linked to sensible ontologies. For example:

http://www.biointerchange.org/gfvo#Quality has an equivalentClass to SIO_000005 (quality). However, neither of these are equivalent (or related) to PATO:0000001

Amino acid is equivalent to SIO_001224. This one is odd because:
1) the definition of the GFVO class is that it “encodes for a Variant_AA or Reference_AA in a GVF file.” However the SIO class that it is equivalent to says that it is “an organic molecule”, which we know is not what is being described with this class (it's an informational representation of said molecule).
2) There is a perfectly reasonable class in SO for this concept: SO:0001237, defined as “A sequence feature that corresponds to a single amino acid residue in a polypeptide”, which is much closer to what I believe you would want the definition to be

There aren’t any logical definitions that I can see. For example, why isn’t the alleleFrequency a cross-product between SO:0001023 (Allele) and PATO:0000044 (frequency).

Most of the ObjectProperties are not reusing existing relationships. For example, there is no need to define hasPart when a perfectly acceptable existing relation is BFO:0000051 (has_part).

In Example 2, I don’t understand the use of an “isAfter” or an “isBefore” relationship between features; I think the intended use is to indicate ordering of objects in a list, not temporal ordering (as might be interpreted from the definition). However, since genomic coordinates would be part of an annotation, qualitative relationships are unnecessary.

Additional comments

In general, I think it was premature to submit this paper for publication. There is not a compelling case for the ontology that was developed (at least, it wasn't compelling in the writing here), nor were there any kind of rigorous use cases and tests of its utility presented.

Reviewer 3 ·

Basic reporting

No Comments

Experimental design

No Comments

Validity of the findings

No Comments

Additional comments

The article is adequately written but is rather short and lacks depth in several areas.

More background information would help a lot. In particular a much better case ought to be made why "yet another ontology" is needed. One or two examples (out of the several provided in suppl. materials) should appear in the main article itself. Turtle syntax is fine but not enough on its own; at least one major example should be more fully explained in the main text with diagram. Similarly the details of how ontology classes/attributes and data structures map to GVFO are documented in detail in the suppl. tables, but this should be explained at a higher level in the main text, ideally with supporting diagram(s). The relationship between GVFO and SIO could also be explained better (what is an upper ontology and what is its purpose?).

Towards the end, the article trails off into a description of RDF (which is a data model, not a file format) and JSON-LD which adds nothing and doesn't help with bringing into focus which problem the ontology is intended to solve.

---

## Round 0.2 · Minor Revisions

· Academic Editor

Minor Revisions

The reviewers point out issues that remain with both the manuscript and the ontology it describes, as well as raise some new issues. I have attempted to clearly state the required revisions based only on issues raised in the first round, and to leave any other responses as optional. If the authors would be so kind as to address this list, I will make an editorial decision without further consulting the reviewers.

1. The introduction sets out no hypothesis. This is still true, but I leave it to the author's discretion how they wish to state their goal, as long as they do so (see point 3).

2. "Generic Feature and Genomic Variation File Formats" section is awkward. This is still true, but I leave this matter of style to the authors' discretion.

3. The authors fail to state the central objective of modeling genomic features in a computable manner using RDF. This is a valid criticism, and I believe the reviewers have pointed out in various ways that the implication in the abstract that GFVO provideds a single "unifying solution to genomic data representation" is overstated given what is reported in the paper. The authors must justify and state their objective more clearly and narrowly.

4. Missing external references to well-defined ontologies, e.g. SIO:cell (SIO_010001). This is valid, though not a flaw in the manuscript, per se. While I recognize that SIO is published separately, the authors may optionally wish to point out where further mappings in GVFO or SIO may be needed, where known.

5. One of the reviewers raises questions about a large number of modeling decisions that were revealed by inspection of the examples. I leave it to the authors' discretion to decide which to correct, which to explain in the manuscript, and which to leave unanswered.

6. "The background and/or discussion still lacks depth with respect to making the case for, and demonstrating the utility of, the new ontology, and a Semantic Web based approach in general" and "don't just provide lots of examples of how genomic variation data can be represented with RDF and GFVO, also show the reader what this is actually good for." This is still valid and closely related to point 3. The authors must add content, however they choose to do so, to rectify this weakness in the manuscript. The authors will likely find it economical to address points 3 and 6 together.

7. "The article skates over and does not mention some of the non-trivial drawbacks of a Semantic Web based approach". This is true, but as this was not raised in the original review, I will leave it to the authors discretion to address or not as they see fit.

8. Inclusion of a Results section (from Editor's comments on prior version). This is still required. Note that the preprint referenced in the authors' response was subsequently published the in the Journal of Biomedical Semantics, not PeerJ, and so not relevant to the requirements of this journal.

Reviewer 2 ·

Basic reporting

As in the previous version, the Introduction is not clearly written to demonstrate how the work fits into the broader field of knowledge. The authors present what feels like a list of facts: there’s lots of genomic data; it lives in files of various formats; xml can encode stuff, like biopax; biopax is also in owl; owl/rdf is useful. This background knowledge would be helpful to provide context for a hypothesis, but there is no hypothesis proposed. Why are you telling me this list of facts? You must have a hypothesis, because you go on to approach solving it by developing a new ontology. Please clearly state your hypothesis before the Methods.

(As copied from previous review since the content of this section has not changed:) I am not sure what the section entitled “Generic Feature and Genomic Variation File Formats” is doing where it is. Is it it’s own section? Is it a sub-section of the Intro? If the latter, it is poor writing style to have a single sub-section.

Experimental design

I appreciate the perceived objective, which I believe to be to model genomic features, from a variety of file formats, in a computable manner using RDF. However, it is not clearly stated.

This ontology is still lacking in the external references to well-defined and used ontologies. For example, the term Cell is defined as gfvo#Cell == SIO:cell (SIO_010001) . in neither ontology is it equivalent to the GO term for cell (GO_0005623) nor the root of the cell ontology (http://www.ontobee.org/browser/rdf.php?o=CL&iri=http://purl.obolibrary.org/obo/CL_0000000 which is used in at least 22 other ontologies) . This is a serious shortcoming considering the broad use of GO for most expression analysis. The authors describe in their response to the first review that there were many classes that could not be mapped as their justification for the need for a new ontology; however, this is an obvious oversight, and not the only example. Justification of the lack of mappings to external ontologies due to the use of SIO is not a good excuse, and perhaps should be reconsidered.

Validity of the findings

Questions about some of the modeling decisions, that need to be clarified in the text:

*For example1:
*why is ExampleSet1 a File? Why not a Set or a Collection?
*why is it "identified" by a BNode with no actual identifier?
*why not have classes for the different file formats, rather than using a string to indicate GFF3, etc?
*can you explain in more detail why you use the :hasAttribute properties to connect the features to a position, rather than just using the faldo:begin/end properties directly? this seems like a lot of extra overhead.

*For example 3:
*GameticPhase is related by two different relationships when comparing ExampleSet3Genotype1 vs 2 (:hasQuality in Genotype1 vs :hasAttribute in Genotype2). Is this intentional? If so, please explain why these are modeled differently.

*For example 5:
*I am still unclear how (and why) the file formats of the source data are encoded. It seems the pattern is to have an identifier with some anonymous node with a version value that is a string. Sometimes this string indicates the format (like in example 1 with “gff-version 3”, but in this example it just says “fileformat 4.2”). Why in the example 1 does it include the pragma value exactly, but in this example it differs (the header has it written as “fileformat=VCFv4.2”).
*Why isn’t the B36 (:Genome) related to the :Reference (taxon)? It seems odd that the Contig/:Landmark would have the attribute of the :Reference (taxon) rather than the :Genome being linked to the taxon. If this makes sense to you, please explain in the article text.

Additional comments

Things that should be addressed somewhere in the manuscript:
*You briefly touch on other genomic file formats in the intro, including FASTA and SAM/BAM, but you should return to how the GVFO does or does not enable the encoding of data from these formats.

*Since you address most of the common file formats, you should address how BED and WIG do or do not fit in this modeling scheme, otherwise it feels intentionally left out.

*Please explain all abbreviations. For example, you have not actually defined GFVO nor SRIQ upon it’s first use in the main body of the text.

*I do not understand why “…only one datatype property was modeled for representing literal values…” What is that one property? Does that mean you only use strings rather than integers? Can you clarify and/or explain this better?

*The text needs to be edited for grammar and active voice. For example, on line 125, “…It is shown…that the comment sections…are adding substantial documentation…” This should have “add” instead of “are adding”. This is one of several grammatical errors related to tense and voice.

*is it a good idea that the data from these formats be transformed into RDF? What does that get the user? Is the resulting TTL/OWL/RDF that is produced of a reasonable size to justify ? Have you computed the size difference in the original vs derived files? I suspect the triple-format to be much larger. Is the size difference worth it? Who would be the user of data converted into this format? The authors should explore these questions in order to demonstrate it’s utility and make the paper more directed to the appropriate audience.

*The conclusions state that they gave supportive evidence demonstrating that the GFVO captures complete G*F and VCF formats. It would be useful if the authors would demonstrate being able to round-trip from one of these formats into the triples, and then into any of the formats once again.

Reviewer 3 ·

Basic reporting

This revised manuscript is an improvement over the original submitted version and addresses some of the issues from my first review. However, the background and/or discussion still lacks depth with respect to making the case for, and demonstrating the utility of, the new ontology, and a Semantic Web based approach in general.

The Introduction briefly describes and highlights shortcomings of a number of existing data representation standards, then proceeds to make some general statements about how the universal RDF triple data model and ontologies has benefits. There's a pretty big, implicit leap here to the conclusion that there's a problem to be solved and that RDF/ontologies are the solution to it.

The key "what is broken and needs to be fixed?" question is not still not asked and answered. One or more real-life user stories would go a long way; i.e. outline some concrete scenarios where an RDF/ontology based approach solves a problem or set of problems that (say) tab-delimited VCF cannot deal with or deals with awkwardly. In other words, don't just provide lots of examples of how genomic variation data can be represented with RDF and GFVO, also show the reader what this is actually good for.

Alternatively (or in addition to the above) how about an exploration of how the Semantic Web approach could be orthogonal to and supplement existing low-level genomic data storage/exchange standards and tools, as a higher level knowledge representation and inference layer.

Finally, the article skates over and does not mention some of the non-trivial drawbacks of a Semantic Web based approach, such as the steep learning curve, re-tooling, performance etc.

Experimental design

--

Validity of the findings

--

---

## Round 0.3 · accepted · Accept

· Academic Editor

Accept

The revisions that have been made are adequate for acceptance.